# Cadinapyridine sesquiterpene alkaloids from *Artemisia annua* and *in vitro* cytotoxicity and antiplasmodial activities

Nicolas Fabre[ID][1]*, Adrien Vitrai[ID][1], Sandra Bourgeade-Delmas[1],
Nathalie Saffon-Merceron[2], Abdou Madjid Olatoundé Amoussa[3], Latifou Lagnika[3],
Agnès Aubouy[1], Valérie Jullian[ID][1]

1 UMR 152 PharmaDev, Université de Toulouse, UPS, IRD, Toulouse, France, 2 Institut de Chimie de Toulouse, ICT UAR 2599, Université Paul Sabatier-Toulouse III, Toulouse, France, 3 Laboratoire de Biochimie et Substances Naturelles Bioactives (LBSNB), Faculté des Sciences et Techniques (FAST), Université d'Abomey Calavi, Abomey Calavi, Bénin

* nicolas.fabre@univ-tlse3.fr

## Abstract

*Artemisia annua* L. (*A. annua*) is a medicinal herb that has been used for the last two millennia to treat various diseases. In African countries, teas prepared from cultivated *A. annua* have been used to treat malaria for the past three decades. In another work, our team investigated the antiplasmodial efficacy of these traditional preparations using a Liquid-Chromatography-Mass Spectrometry metabolomic approach on various teas prepared from different batches of *A. annua* collected in Benin and in France and highlighted an original nitrogenous compound. The present work aimed to isolate and characterize its structure. Therefore we describe the fractionation of an alkaloid extract from a Beninese sample of the plant's aerial parts led to the isolation of two previously undescribed alkaloids, annuanine A (**1**) and annuanine B (**2**) along with the known fabianine (**3**), all possessing a very unusual skeleton. Their structures were determined through NMR and MS data analyses. The structure of **1** was further confirmed by single-crystal X-ray structural analysis. Compounds **1** and **2** were evaluated for cell-growth inhibition on Caco-2, VERO, and Thp1 cells, as well as on the *Plasmodium falciparum* FcB1 strain. Unfortunately, the two compounds **1** and **2** were inactive in these *in vitro* models. Comparison of LC-HRMS data between annuanine B and *A. annua* tea allowed us to identify it as the nitrogenous compound highlighted by our previous study. These results enhance the chemical knowledge of this well-known ethnopharmacological herb and highlight a very rare alkaloid skeleton that may have formed from the degradation of artemisinin during plant storage.

**Data availability statement:** All NMR spectra and UHPLC/HRMS data [and related documentations] that support the findings of this study are openly available in DataSuds repository (IRD, France) at https://doi.org/10.23708/YYQHFP. Data reuse is granted under CC-BY license.

**Funding:** The author(s) received no specific funding for this work.

**Competing interests:** The authors have declared that no competing interests exist.

## Introduction

*Artemisia annua* L. (sweet wormwood, annua wormwood, or Chinese wormwood) has been recognized as an important ethnomedicinal herb for the last two millennia; its mention appears in the "recipes for 52 kinds of diseases" found in the Mawangdui Han dynasty tomb dating from 168 B.C., where the herb is recommended for treating hemorrhoids [1]. Since then, it has been used by traditional Chinese medicine herbalists to treat various diseases, including intermittent fevers due to malaria, tuberculosis, lice, lingering heat in joints and bones, wounds, scabies and dysentery [2–4]. Following the discovery of the seco-cadinane sesquiterpene lactone endoperoxide artemisinin [5], the active antimalarial principle of *A. annua* by Youyou Tu in 1971 [6], more than 600 phytochemical constituents have been isolated from this species [7–10]. These constituents can be categorized into three main groups: terpenoids with varying levels of oxidation, aromatic hydrocarbons like phenylpropanoids, and flavonoids. The latter group constitutes the second largest category after terpenoids. Among the terpenoids, sesquiterpenes are the most abundant and diverse group of natural products found in *A. annua* [10]. Various acyclic, monocyclic, bicyclic and tricyclic sesquiterpenoids are present in the plant; however, the largest class consists of amorphane/cadinane six-membered bicyclic sesquiterpenes [9–11]. Regarding nitrogen-containing derivatives, some peptides and other nitrogenous natural products (aurantinamide acetate, hydroxypurine derivative, benzothiazole) have been reported [12–14].

A recent study conducted by our team focused on investigating the antimalarial efficacy of traditional preparations of Chinese wormwood using untargeted metabolomic approach conducted on various teas prepared from different batches of *A. annua* collected in Benin and France. The phytochemical compositions, combined with *in vitro* $IC_{50}$ values on *Plasmodium falciparum* of different tea samples revealed a feature with an original molecular formula, which appears to be linked to the best antimalarial activity, and which content was increasing during the time of storage of the plant [15]. In the present work, we describe the structural elucidation and preliminary biological properties of this new derivative, along with two other structurally related compounds, including a known derivative. The biosynthetic origin of these secondary metabolites is also discussed.

## Materials and methods

### Plant materials

Leaves and stems of *A. annua* L. (Asteraceae) were cultivated and collected in a plantation in Banigbe (6° 48′ 58″ N, 2° 39′ 02″ E) in the Plateau region of Benin in December 2021. The species was identified by Prof. Hounnankpon Yedomonhan, head of the National Herbarium of Benin (University of Abomey-Calavi). A voucher specimen was deposited in this institution under the register No. A0012 A/HNB.

### Chemicals

Acetone (≥99%) and ethanol (96%) were obtained from VWR International. Chloroform, toluene and methanol (analytical or HPLC grade) were purchased from Fisher

Scientific SAS. Ammonium hydroxide (28–30%) was obtained from ThermoFisher Scientific. Hydrochloric acid (37%) was from Sigma-Aldrich. Dragendorff reagent was prepared as follows: Solution A: a mixture of 0.85 g of bismuth nitrate (98%, Merck) with 40 mL of distilled water and 10 mL of acetic acid (99.8–100.5%, NORMAPUR® from VWR International). Solution B: 8 g of potassium iodide (Rectapur from VWR International) were dissolved in 20 mL of distilled water. Final solution: 5 mL of Solution A, 5 mL of Solution B, 9.43 mL of acetic acid, and 100 mL of distilled water.

## Extraction and purification of alkaloids

Obtaining total alkaloids: 650 g of dried leaves and stems were moistened with 650 mL of ammonium hydroxide 28–30%/ $H_2O$ (50/50, v/v). After 10 minutes, the plant material was extracted by reflux with 6 L of boiling chloroform during 2 hours with mechanical stirring. The mixture was then filtered through a Büchner funnel. The filtrate was then treated liter by liter. One liter was transferred to a separating funnel and washed three times with 300 mL of a 2% (m/v) HCl aqueous solution. Seventy-five mL of 28–30% ammonium hydroxide were added to basify the aqueous solution to a pH of 10 (monitored with pH paper). The basic aqueous phase was then extracted 3 times with 300 mL of chloroform. The solvent was then evaporated yielding 600 mg of an orange/brown viscous oil called total alkaloid. The total alkaloid (200 mg) was fractionated on a silica PuriFlash® SI-HC 25 μm, 25 g silica cartridge (Interchim). The total alkaloid was injected using a dry-load cartridge mixed with 1.5 g of celite and the cartridge was then filled with sand. The column was eluted using ammonium acetate (5 M)/ethanol/toluene/acetone (1/3/20/20) at a flow rate of 15 mL/min. TLC monitoring with Dragendorff reagent led to 6 fractions. After repeating this fractionation in order to treat the whole total alkaloid (600 mg), the quantity obtained for the 6 fractions was as follows: F0 (6 mg), F1 (115 mg), F2 (68 mg), F3 (52 mg), F4 (25 mg), F5 (41 mg), and a F6 fraction obtained after a methanol washing of the column. Fraction F1 (27 mg) was treated by semi-preparative HPLC after solubilization in methanol (15 mg/mL) and six successive runs. An isocratic mode of 70% distilled water and 30% methanol with a 3 mL/min flow rate was used for the first 15 minutes, followed by a column washing step with 95% methanol for the last 15 minutes. After regrouping subfractions, evaporation of methanol and lyophilization of water, 2 mg of compound **3** (F1-4) were obtained. Fraction F4 (25 mg) was treated by semi-preparative HPLC after solubilization in methanol (12 mg/ mL) and injected in seven successive runs. A gradient of distilled water with 0.1% formic acid (A) and methanol with 0.1% formic acid (B) at a 15 mL/min flow rate was used, and consisted of 5% B from 0 to 1 minute, 5–30% B from 1 to 20 minutes, 30–95% B from 20 to 30 minutes, and 95% B from 30 to 35 minutes. After regrouping subfractions, evaporation of methanol, and lyophilization of water, 2 mg of compound **1** (F4-1) were obtained. Fraction F5 (36 mg) was treated by MPLC using a Puriflash® XS 520 Plus Interchim system. The F5 fraction was solubilized in 3 mL of distilled water/methanol (1/1) and injected at the top of a C18 silica cartridge (RS 15 C18 (CHROMABOND Flash, MACHEREY-NAGEL)). An elution gradient of distilled water (A)/methanol (B) consisting of 0% B from 0 to 3 minutes, 0–95% B from 3 to 40 minutes, and 95% B from 40 to 45 minutes was used at a 15 mL/min flow rate. TLC monitoring with Dragendorff led to the grouping of tubes. After evaporation of MeOH and lyophilization of water, 16 mg of pure compound **2** (PF1-F5-1) were obtained.

Compound **1**: colorless needle-shaped crystals, UV (methanol) $\lambda_{276}$ (log ε) 3.430; $[\alpha]^{20}_D$ - 99.7 (c 0.125; methanol); $^1H$ and $^{13}C$ NMR data (deuterated chloroform ($CDCl_3$), 500 MHz), see Table 1; high resolution electrospray ionization mass spectrometry (HRESIMS) m/z 233.16434 [M+H]⁺ Δppm = −2.144 (calcd for $C_{14}H_{21}ON_2^+$, 233.1653).

Compound **2**: yellowish film, UV (methanol) $\lambda_{278}$ (log ε) 4.054; $[\alpha]^{20}_D$ + 35.4 (c 0.766; methanol; $^1H$ and $^{13}C$ NMR data ($CDCl_3$, 500 MHz), see Table 1; HRESIMS m/z 234.14873 [M+H]⁺ Δppm = −0.536 (calcd for $C_{14}H_{20}O_2N^+$, 234.1494).

Fabianine (**3**): white film, UV (methanol) $\lambda_{275}$ (log ε) 2.678; $[\alpha]^{20}_D$ - 17.5 (c 0.150; methanol); $^1H$ and $^{13}C$ NMR data ($CDCl_3$, 500 MHz), see Table 1; HRESIMS m/z 220.16907 [M+H]⁺ Δppm = −2.366 (calcd for $C_{14}H_{22}ON^+$, 220.1701).

## General analysis and purification equipment

Compounds were purified on silica cartridges (PuriFlash® SI-HC 25 μm, 25 g) using a Puriflash® XS 520 Plus preparative Medium Pressure Liquid Chromatography MPLC Interchim system equipped with a diode detector operating at

200−400 nm and semi-preparative high performance liquid chromatography (HPLC) using a HITACHI LaChrom system (MERCK) consisting of a quaternary LaChrom L-7100 pump and a LaChrom L-7455 photodiode array detector, and a Luna C18 100 Å column (10×250 mm, 5 μm) (Phenomenex). Solvent was removed from the samples using a rotary evaporator (KNF RC 600) equipped with a water bath and a SC 920 G vacuum pump. Samples were freeze-dried using a Labconco FreeZone 2.5L benchtop freeze dryer set at −80°C. Ultaviolet-visible (UV-Vis) absorbance spectra were recorded on a Specord 205 spectrophotometer (Analytik Jena, Jena, Germany) using quartz cuvettes with a 1 cm path length. Optical rotations were determined using a JASCO P2000 digital polarimeter. Nuclear Magnetic Resonance (NMR) spectra were recorded on a Bruker Avance 500 MHz instrument equipped with a 5 mm TCI Prodigy CryoProbe, using tetramethylsilane (TMS) as reference. Spectrometers were controlled and analyzed using Topspin and MNova softwares. Ultra-high pressure liquid chromatography coupled to diode-array detector and high-resolution mass spectrometry (UHPLC/DAD/HRMS) analyses were performed using an UHPLC Ultimate 3000 system (Dionex) equipped with an LTQ-Orbitrap XL mass spectrometer (Thermo Fisher Scientific), a DAD system (Dionex), and a Waters Acquity PREMIER C18 UPLC BEH 100 Å column (2.1×100 mm, 1.7 μm). Chromeleon Xpress 6.8 (Dionex) and Xcalibur 3.0 (Thermo Fischer Scientific) softwares were used for data acquisition and analysis. Analytical thin-layer chromatography (TLC) was carried out on precoated silica gel plates (Merck, Kieselgel 60 F254, 0.25 mm); detection was performed under UV 254 nm and UV 366 nm and after spraying with Dragendorff reagent.

## X-ray crystallography

Compound **1** was crystallized by slow evaporation form methanol/$H_2O$ (1/1) at room temperature. Crystal data were collected at 193 K using Cu Kα radiation (wavelength = 1.54178 Å) on a Bruker AXS D8-Venture equipped with a Photon III diffractometer using a 30 W air-cooled microfocus source (ImS) with focusing multilayer optics. Phi- and omega-scans were used. Space group was determined on the basis of systematic absences and intensity statistics. Semi-empirical absorption correction was employed. [16] The structure was solved using an intrinsic phasing method (ShelXT). [17] All non-hydrogen atoms were refined anisotropically using the least-square method on F2. [18] Hydrogen atoms were refined isotropically at calculated positions using a riding model with their isotropic displacement parameters constrained to be equal to 1.5 times the equivalent isotropic displacement parameters of their pivot atoms for terminal sp3 carbon and 1.2 times for all other carbon atoms. H atoms on N2 were located by difference Fourier Map. The ketone and $NH_2$ groups were found to be disordered. The SADI restraint and equal xyz and Uij constraints (EXYZ, EADP) were applied to refine some moieties of the molecule and to avoid the collapse of the structure during the least-squares refinement by the large anisotropic displacement parameters.

CCDC-2401400 contains the supplementary crystallographic data for this paper. These data can be obtained free of charge from the Cambridge Crystallographic Data Centre via https://www.ccdc.cam.ac.uk/structures/.

Selected data for **1**: $C_{14}H_{20}N_2O$, $M$ = 232.32, orthorhombic, space group $P2_12_12_1$, $a$ = 5.0758(3) Å, $b$ = 15.9065(8) Å, $c$ = 16.0856(8) Å, $V$ = 1303.13(12) Å$^3$, $Z$ = 4, crystal size 0.25 x 0.08 x 0.07 mm$^3$, 32729 reflections collected (2295 independent, R$int$ = 0.0558), 170 parameters, 2 restraints, $R$1 [I > 2σ(I)] = 0.0428, $wR$2 [all data] = 0.0897, absolute structure parameter: 014(14), largest diff. peak and hole: 0.118 and −0.147 eÅ$^{-3}$.

## Evaluation of *in vitro* antimalarial activity

The *in vitro* anti-malarial activity was investigated using the SYBR Green I-based fluorescence assay as described by Smilkstein and coworkers with slight modifications [19]. Briefly, the asexual intra-erythrocytic stage of *P. falciparum* laboratory strain FcB1 (chloroquine-resistant strain) was maintained in RPMI 1640 medium containing l-glutamine 200 M, 4-(2-hydroxyethyl)-1-piperazine ethane sulfonic acid (Hepes) 25 mM and 5% human serum (Etablissement Français du Sang—EFS, Toulouse, France). For anti-malarial drug assays, stock solutions of compounds were diluted serially in Roswell Park memorial institute (RPMI) 1640 culture medium in a 96-well plate. A suspension of sorbitol-synchronized,

infected red blood cells (iRBCs) was adjusted to 1% parasitemia and 2% hematocrit in complete medium and added to the wells. Negative controls were prepared with a suspension of iRBCs. Chloroquine was used as the positive control. Test plates were incubated at 37°C, 5% $CO_2$ for 48 h. Afterwards, 100 µL SYBR Green I fluorescent lysis buffer were added to each well and incubated in a dark place at room temperature for 2 h. Fluorescence data were acquired using a fluorescence plate reader (BMG Fluostar Galaxy Labtech) with excitation and emission wavelengths at 485 nm and 518 nm, respectively. The fluorescence values (after subtraction of the background fluorescence of the non-parasitized RBCs) were plotted against the log of the drug concentration, and analyzed by non-linear regression (sigmoidal dose response/ variable slope equation) to yield the 50% inhibition concentration ($IC_{50}$) that served as a measure of the anti-malarial activity.

### Evaluation of cytotoxicity on Vero cell line

The evaluation of cytotoxicity on the Vero cell line was conducted using the 3-[4,5-dimethylthiazol-2-yl]-2,5 diphenyl tetrazolium bromide (MTT) assay described by Mosmann [20], with modifications as follows. Briefly, cells ($5.10^4$ cells/ mL) in 100 µL of complete medium, [Dubelcco's modified eagle medium (MEM) supplemented with 10% fetal calf serum (FCS), 2 mM L-glutamine and antibiotics (100 U/mL penicillin and 100 µg/mL streptomycin)] were seeded into each well of 96-well plates and incubated at 37°C in a humidified, 5% $CO_2$ with 95% air atmosphere. After a 24 h incubation, 100 µL of medium with various concentrations of tested compounds and appropriate controls were added and the plates were incubated for 72 h at 37°C. Each plate-well was then microscope-examined for detecting possible precipitate formation before the medium was aspirated from the wells. One hundred µL of methyl thiazolyl tetrazolium (MTT) solution (0.5 mg/mL in complete medium) were then added to each well. Cells were incubated for 1 h at 37°C. After this time, the MTT solution was removed and dimethylsulfoxide (DMSO) (100 µL/well) was added to dissolve the resulting formazan crystals. Plates were shaken vigorously (300 rpm) for 5 min. The absorbance was measured at 570 nm with a microplate spectrophotometer (BIOTEK Eon). $CC_{50}$ were calculated by non-linear regression analysis processed on dose–response curves, using TableCurve 2D V5 software.

### Evaluation of cytotoxicity on Caco-2 cells

The cytotoxicity on Caco-2 cells was assessed using the MTT assay [20]. Caco-2 cells ($1.10^5$ cells/mL) in 100 µL of complete medium [DMEM High Glucose supplemented with 10% FCS, 2 mM L-glutamine, antibiotics (100 U/mL penicillin, 100 µg/mL streptomycin) and non-essential amino acids 1X solution (1X NEAA)] were seeded into each well of 96-well plates and incubated at 37°C in a humidified, 5% $CO_2$ with 95% air atmosphere. After a 24 h incubation, various concentrations of molecules and appropriate controls were then added (100 µL) and the plates were incubated for 72 h at 37°C. Each well was then microscope-examined for detecting possible precipitate formation before aspiration of the medium. MTT solution (0.5 mg/mL in complete medium, 100 µL) were then added to each well. Cells were incubated for 2 h at 37°C. The MTT solution was then removed and DMSO (100 µL) was added to dissolve the resulting formazan crystals. Plates were shaken vigorously (300 rpm) for 5 min. The absorbance was measured at 570 nm with a microplate spectrophotometer (BIOTEK Eon). Water and DMSO were used as blank and doxorubicin (Sigma Aldrich) as positive control. Fifty percent cytotoxic concentration ($CC_{50}$) were calculated by non-linear regression analysis processed on dose–response curves, using TableCurve 2D V5 software.

### Evaluation of cytotoxicity on the Thp1 cells

The MTT assay was used to evaluate the cytotoxicity of compounds 1–2 against human monocytes Thp1 cells [20]. Thp1 cells ($0.77.10^5$ cells/mL) in 200 µL of complete medium [RPMI 1640 supplemented with 10% FCS, 2 mM L-glutamine and antibiotics (100 U/mL penicillin and 100 µg/mL streptomycin)] + phenylmethoxypenicillin (50 ng/mL)] were seeded into each well of 96-well plates and incubated at 37°C in a humidified, 5% $CO_2$ with 95% air atmosphere. After a 96-h

incubation, plates were rinse 3 times with medium and 100 µL of medium were added. One hundred µL of medium with various compounds and appropriate controls were added and the plates were incubated for 72 h at 37°C. Each plate-well was then microscope-examined for detecting possible precipitate formation before the medium was aspirated from the wells. One hundred µL of MTT solution (0.5 mg/mL in complete medium) were then added to each well. Cells were incubated for 2 h at 37°C. After this time, the MTT solution was removed and DMSO (100 µL) was added to dissolve the resulting formazan crystals. Plates were shaken vigorously (300 rpm) for 5 min. The absorbance was measured at 570 nm with a microplate spectrophotometer (BIOTEK Eon). $CC_{50}$ were calculated by non-linear regression analysis processed on dose–response curves, using TableCurve 2D V5 software.

## Results and discussion

### Structure of compound 1

Compound **1** was obtained as colorless needle-shaped crystals. Its molecular formula was determined to be $C_{14}H_{20}N_2O$ based on the positive HRESIMS $m/z$ 233.1644 [M + H]+ (calc for $C_{14}H_{21}N_2O$, 233.1648, Δ 2.9 ppm), indicating six degrees of unsaturation. The MS/MS spectrum (S1 Fig in S1 File) of $m/z$ 233 leads to $m/z$ 216 as the only ion fragment, suggesting a primary amine (loss of $NH_3$) in the structure of **1**. A UV spectrum with a $\lambda_{max}$ of 279 nm indicated the presence of an aromatic ring in the structure of compound **1**. Inspection of the $J$-modulated $^{13}$C-NMR (Table 1 and S3 Fig in S1 File) and HSQC spectroscopic data (S4 Fig in S1 File) revealed fourteen carbon resonances, indicating the presence of three methyl groups, two methylene groups, five methines (including two aromatics), and four quaternary carbons divided in three aromatic carbons and one amide carbonyl ($\delta_c$ 178.5). Notably, only five aromatic carbon resonances appeared in the $^{13}$C-NMR spectrum ($\delta_c$ 135.8, 136.1, 121.2, 154.4, 156.8 for C-1, C-2, C-3, C-4 and C-6, respectively), suggesting the presence of a pyridine moiety in the structure of **1** [21,22]. Investigation of $^1$H-NMR (Table 1 and S2 Fig in S1 File) and

**Table 1. $^1$H and $^{13}$C NMR data for compounds 1–3 (in $CDCl_3$).**

| no. | 1 $\delta_c{}^a$, type | 1 $\delta_H{}^b$ (mult, $J$, Hz) | 2 $\delta_c{}^a$, type | 2 $\delta_H{}^b$ (mult, $J$, Hz) | 3 $\delta_c{}^a$, type | 3 $\delta_H{}^b$ (mult, $J$, Hz) |
|---|---|---|---|---|---|---|
| 1 | 135.8 C | | 137.4 C | | 135.5 C | |
| 2 | 136.1 CH | 7.48 d (7.9) | 138.0 CH | 7.68 dd (8.0, 1.0) | 135.8 CH | 7.50 d (8.0) |
| 3 | 121.2 CH | 6.97 d (7.9) | 122.6 CH | 7.12 d (8.0) | 121.1 CH | 6.97 d (8.0) |
| 4 | 154.4 C | | 152.6 C | | 153.6 C | |
| 5 | – | – | – | – | – | – |
| 6 | 156.8 C | | 154.6 C | | 158.5 C | |
| 7 | 44.9 CH | 3.11 m ax | 40.6 CH | 3.46 dd (11.8, 6.0)/ 3.48 dd (12.3, 5.6)* | 50.6 CH | 3.01 dd (11.7, 6.4) |
| 8 | 25.6 CH$_2$ | 1.65 m 8β ax<br>2.13 dddd (13.8, 5.7, 5.7, 3.0) 8α eq | 29.1 CH$_2$ | 1.67 m 8β ax<br>2.02 m 8α eq (overlap) | 26.3 CH$_2$ | 1.45 m 8β ax<br>2.13 m 8α eq |
| 9 | 31.6 CH$_2$ | 1.37 dddd (13.0, 13.0, 10.5, 2.8) 9α ax<br>2.01 dddd (13.4, 5.4, 5.4, 2.9) 9β eq | 31.8 CH$_2$ | 1.41 m9α ax<br>2.02 m 9β eq (overlap) | 32.1 CH$_2$ | 1.35 9α ax<br>1.98 m 9β eq |
| 10 | 32.6 CH | 2.81 m | 32.4 CH | 2.81 m (overlap)/ 2.81 dq (12.2, 6.2) ax* | 32.8 CH | 2.77 m |
| 11 | 43.1 CH | 3.46 qd (7.2, 2.7) | 47.8 CH | 2.81 m (overlap)/ 2.73 qd (7.5, 1.3)* | 74.3 C | |
| 12 | 14.8 CH$_3$ | 1.17 d (7.2) | 12.8 CH$_3$ | 1.00 d (7.5) | 29.1 CH$_3$ | 1.29 s |
| 13 | 178.5 C | | 178.9 C | | 24.3 CH$_3$ | 0.95 s |
| 14 | 21.5 CH$_3$ | 1.26 d (6.9) | 20.8 CH$_3$ | 1.29 d (6.8) | 21.1 CH$_3$ | 1.27 d (6.9) |
| 15 | 24.1 CH$_3$ | 2.49 s | 21.9 CH$_3$ | 2.55 s | 23.1 CH$_3$ | 2.46 s |

[a]Measured at 125 MHz. [b]Measured at 500 MHz. δ in ppm. * Signal obtained when the spectrum is recorded in $CD_2Cl_2$.

HSQC spectra (S4 Fig in S1 File) confirmed the presence of three methyl groups (two doublets at $\delta_H$ 1.17, 1.26 and a deshielded singlet at $\delta_H$ 2.49), two cyclic methylenes (well resolved AB systems between $\delta_H$ 1.37 and 2.13) and two vicinal aromatic protons (doublets with coupling constants of 7.9 Hz at $\delta_H$ 6.97 and 7.48), indicating a trisubstituted pyridine. Lastly, three methine signals were observed, one as a quartet (H-11 at $\delta_H$ 3.46) and two cyclic methines appearing as multiplets (H-7 and H-10 at $\delta_H$ 3.11 and 2.81, respectively). With six degrees of unsaturation, compound **1** apparently contained a six-membered four substituted ring alongside the pyridine ring and the amide carbonyl (Fig 1). The planar structure of **1** was drawn according to key HMBC and COSY (S5 Fig and S6 Fig in S1 File, respectively) cross peaks depicted in Fig 2.

The relative configuration of **1** can be deduced from an analysis of the NOE spectrum (S7 Fig in S1 File) and the coupling constants observed for H-8α, H-9α, H-9β, H-10 in the ¹H-NMR spectrum. First, NOESY correlations between H-10 (2.81 ppm) and H-8β (1.65 ppm), as well as between H-7 (3.11 ppm) and H-9α (1.37 ppm), support an axial orientation for these four hydrogens. The axial position of H-9α is confirmed by its multiplicity, which appears as a dddd, with three large coupling constants ($J_{9\alpha,9\beta}/J_{9\alpha,8\beta}/J_{9\alpha,10ax}$), thus confirming an axial position for H-10 as well. The axial orientation of both H-7

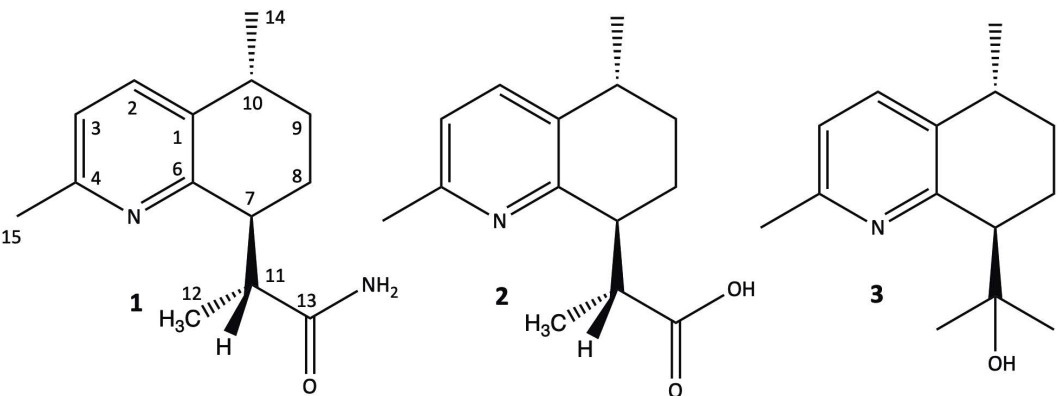

**Fig 1. Structures of cadinapyridines 1-3.**

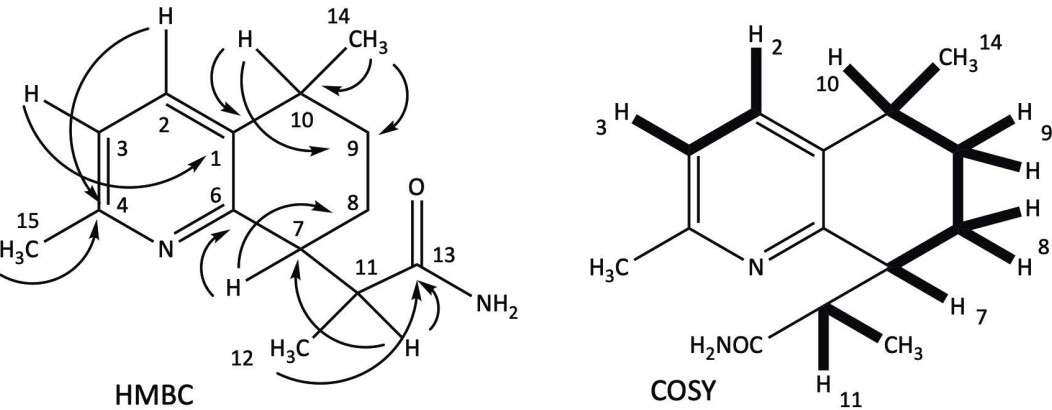

**Fig 2. Key HMBC and COSY correlations of 1.**

and H-10 indicates a *trans* relative configuration for the six-membered ring. These deductions were confirmed by X-ray crystallography (molecular view depicted in Fig 3), which also allowed for the determination of the absolute configurations 7$S$, 10$R$, and 11$R$ at the chiral centers of **1**, with a Flack parameter of 0.14 (14). Consequently, the structure of **1** was defined as (11$R$)-[(7$S$,10$R$)-4,10-dimethyl-7,8,9,10-tetrahydroquinolin-7-yl]propanamide that we call annuanine A.

## Structure of compound 2

Compound **2** was obtained as a yellowish film. The molecular formula was determined to be $C_{14}H_{19}NO_2$ based on the molecular ion at *m/z* 234.1492 [M + H]$^+$ (calcd for $C_{14}H_{20}NO_2$, 234.1489, Δ 1.6 ppm), indicating six degrees of unsaturation. Compared to compound **1**, the molecular formula of **2** shows that NH has been replaced by an oxygen atom ($C_{14}H_{20}N_2O$ for **1**). This observation suggests that the amide function of **1** disappeared in favor of a carboxylic function in **2**, justifying the difference of one mass unit between the two compounds. This assumption is supported by the MS$^2$ of *m/z* 234 leading to *m/z* 216 (loss of $H_2O$) and MS$^3$ of 234 > 216 leading to *m/z* 188, implying a loss of carbonyl (S8 Fig in S1 File). The $^{13}$C NMR spectrum of **2** (Table 1, S9 Fig in S1 File) was almost superimposable to that of compound **1**. Table 1 also reports slight differences in the $^1$H NMR spectrum of **2**. In particular, the chemical shift of H-7 appeared at a lower field ($\delta_H$ 3.46), while H-11 signal was present at a higher field ($\delta_H$ 2.81 instead of $\delta_H$ 3.46 for **1**). Consequently, certain signals were overlapped. The 2D HMBC and COSY spectra of **2** (S12 Fig and S13 Fig in S1 File), identical to those of **1**, established the planar structure of this new compound depicted in Fig 1.

The relative configuration of the chiral centers of **2** was investigated based on NOESY experiment (S14 Fig in S1 File) and $^1$H NMR coupling constants. Due to overlaps observed in the $^1$H NMR spectrum of **2** recorded in CDCl$_3$, NMR spectra recorded in CD$_2$Cl$_2$ allowed for the separation of signals for the key hydrogens H-10 and H-11, whose resonances are now localized at $\delta_H$ 2.85 and 2.77, respectively (Fig 4). Compared to compound **1**, H-11 in **2** appeared as a quartet doublet ($J_{11,H3-12}/J_{11-7}$ = 7.5/1.3 Hz) similar to **1**, while the multiplicity of H-7 in **2** is a double doublet ($J_{7,8\beta}/J_{7,8\alpha}$ = 12.3/5.6 Hz) as depicted in Fig 4, indicating an axial orientation for H-7 and H8β as in compound **1**. The same conclusion can be drawn for H-10, which appears as a doublet of quartet ($J_{10,9\alpha}/J_{10,H3-14}$ = 12.2/6.2 Hz), confirming its axial position. These findings lead to the conclusion that compound **2** has the

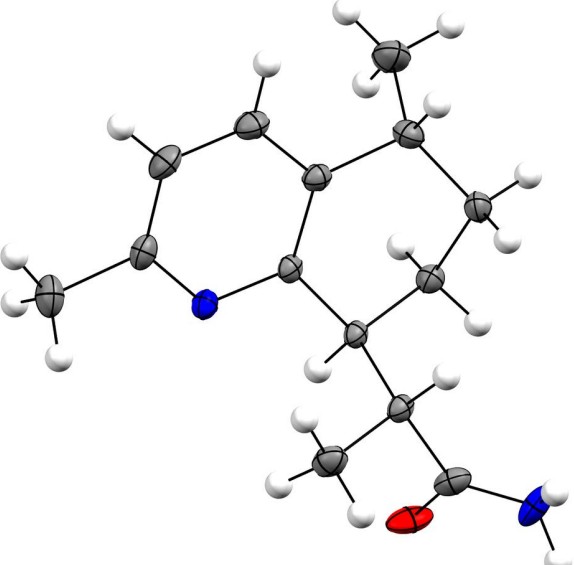

**Fig 3. Molecular view of compound 1 based on single X-ray crystallographic analysis.** Thermal ellipsoids represent 30% probability. Disordered atoms (O and NH$_2$) are omitted for clarity.

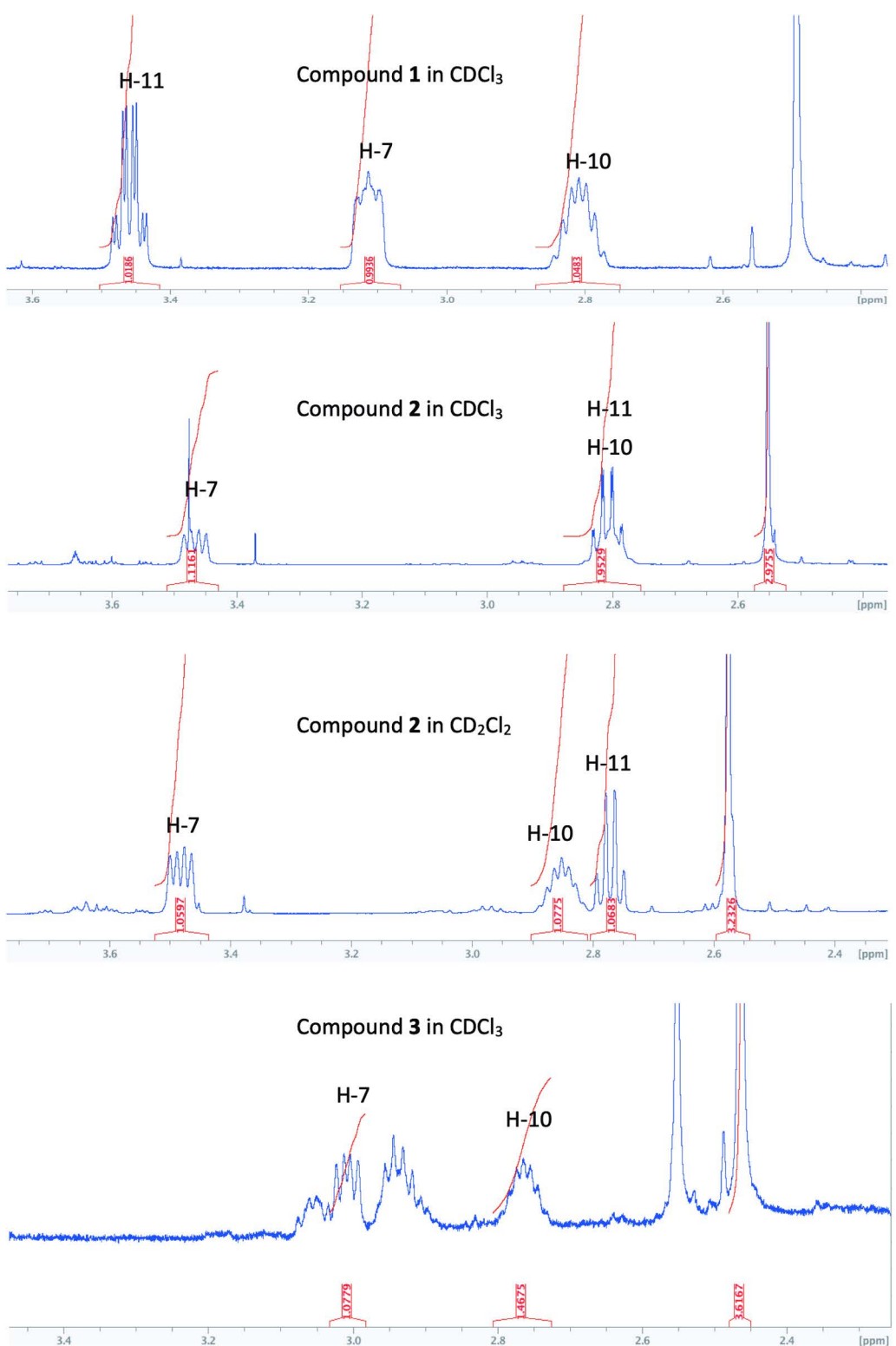

**Fig 4.** ¹HNMR multiplicities of H-7, H-10 and H-11 for cadinapyridines 1-3.

same configuration (7*S*, 10*R*, and 11*R*) for the chiral centers as compound **1**. Therefore, the structure of **2** was determined as (11*R*)-[(7*S*,10*R*)-4,10-dimethyl-7,8,9,10-tetrahydroquinolin-7-yl] propananoic acid and called annuanine B.

### Structure of compound 3

Compound **3** was obtained as a white film and has the molecular formula $C_{14}H_{21}NO$ according to its HRESIMS data ([M + H]⁺ at *m/z* 220.1689 for $C_{14}H_{22}NO$, Δ 0.68 ppm) (S15 Fig in S1 File). All ¹H and ¹³C NMR data were in good agreement with those of Fabianine [23], an alkaloid first isolated from *Fabiana imbricata* by Edwards and Elmore in 1962. [24] Therefore, the configuration at C-7 seemed to be unclear in the few articles published on the Fabianine structure [23 25]. Here, the stereochemistry was deduced on analogy to **1** and **2** since H-7 and H-10 signals in the ¹H NMR spectrum of Fabianine (**3**) were very similar, as depicted in the Fig 4.

### Biosynthetic origin

To our knowledge, compounds **1** and **2** are the second and third representatives, after fabianine (**3**), of this type of alkaloids. Examining their structure reveals a cadinane-type sesquiterpene skeleton in which a carbon atom has been replaced by a nitrogen, leading to a 2-methylpyridine moiety. Interestingly, other sesquiterpene pyridine alkaloids, involving a guaiane core, have been described in the *Artemisia* genus. These are guaipyridine sesquiterpene alkaloids known as rupestines [21,22]. We propose the term cadinapyridine sesquiterpene alkaloids for this family of natural compounds. The two structural groups can be easily differentiated based on their ¹³C NMR spectra, as guaipyridines contain fifteen carbons in their basic structure, whereas cadinapyridines have only fourteen. The biogenesis of guaipyridines was proposed by Zhang and coworkers [26] to originate from a basic guaiane skeleton (formed from farnesyl pyrophosphate), where a cleavage of a C = C double bond produces a seco-regular sesquiterpene named xanthane. The inclusion of a nitrogen atom between the two carbonyl groups leads to the guaipyridine framework (Fig 5A). Schmeda-Hirschmann and Papastergiou [23] proposed a similar biosynthetic pathway for fabianine, involving the additional loss of one carbon, supported by their isolation of the C-11 OH cadinane sesquiterpene and C-11 OH seco-sesquiterpene precursor of fabianine (Fig 5B). A similar pathway can be proposed for the origin of compound **2**, starting from dihydroartemisinic acid, the well-known precursor of artemisinin, and leading to an oxidized intermediate, arteannuin Q, previously isolated from *A. annua* (Fig 5C) [27]. Notably, a series of synthetic cadinapyridines (including compound **2** as an intermediate) has been described to design novel transmembrane receptor smoothened (Smo) antagonists involved in the genesis of certain cancers [28]. This series was constructed from the acidic degradation of artemisinin, yielding a diketo seco-cadinane intermediate, which is the methyl ester of arteannuin Q shown in Fig 5C. The latter, treated with $NH_4AcO$ and $Cu(AcO)_2$ at room temperature, was easily transformed into a pyridine-fused bicyclic framework (a cadinapyridine sesquiterpene alkaloid), serving as the starting point for the synthetic series of Smo antagonists [28]. This intriguing coincidence with our work raises questions about the formation of compounds **1**–**3**. Are they of biosynthetic origin, or do they constitute degradation products of artemisinin and/or other cadinane sesquiterpenoids present in *Artemisia annua*? Due to the experimental conditions of alkaloid extraction, we cannot rule out that artemisinin (or another molecule present in *A. annua*) was transformed in compounds **1**–**3** during the extraction process. Comparison of chromatograms obtained from pure compound **2** and *Artemisia* tea, under the same LC-HRMS conditions, allowed us to identify the alkaloid present in the tea as compound **2** with a good level of confidence (comparable retention time, HRMS and HRMS/MS signals, S21 Fig in S1 File). In our previous work, we noted that the intensity of its LC-HRMS signal increased with the duration of storage of the plant used for the tea preparation, while the content of artemisinin decreased [15]. We also present here additional data showing that compound **2** signal was absent in a tea made from fresh *A. annua*, and started to appear after the drying process (S23 and S24 Figs in S1 File). This reinforces the hypothesis that **2,** and possibly arteannuin Q, are likely degradation products of artemisinin (or another molecule present in *A. annua*) during prolonged storage of the plant organs. Regarding the other two compounds, **1** was not detected in the teas, and maybe a purification artifact. However, the obtention of its X-Ray structure

**Fig 5. Proposed biosynthetic pathways of guaipyridine (A, according to [26]), cadinapyridine (B, according to [23]) sesquiterpene alkaloids, and for compound 2 (C).**

was necessary to reinforce the demonstration of the structure of compound **2** especially for the 3D-structure. Compound **3** was not observed in the LC-MS profiles until it was identified in the present study as a trace.

## Biological evaluation

As annuanine B (compound **2**) was first highlighted in *A. annua* teas following a metabolomic analysis of several samples revealing significant differences in their *in vitro* antimalarial activities [15], annuanine A (**1**) and annuanine B (**2**) were assessed for their bioactivities against the chloroquine-resistant *Plasmodium falciparum* FcB1 strain. To evaluate potential selectivity for activity against *P. falciparum* or cancer cells, both compounds were also tested for their ability to inhibit cell growth in human and non-human healthy cells (human monocytes Thp1 and monkey kidney epithelial Vero), as well as in a cancer cell line (human colorectal adenocarcinoma Caco-2). As shown in S1 Table in S1 File, none of these compounds showed any activity in these models ($CI_{50} > 427$ μM against *P. falciparum* and > 214 μM against the other cell lines). To further understand the statistical results obtained from the metabolomic investigation, particularly regarding the discrimination of compound **2** as responsible for the most interesting antimalarial activity of *A. annua* teas, synergy studies involving

**2** and artemisinin have been undertaken (not shown). Unfortunately, the antimalarial activity observed with different concentrations mixing the two compounds was exclusively attributable to the reference antimalarial artemisinin. Thus, the link between compound **2** and antimalarial activity highlighted by the metabolomic study may be the result of a correlation between the quantity of artemisinin and the quantity of compound **2**.

## Conclusion

In this study, we isolated and characterized two previously undescribed cadinapyridine sesquiterpene alkaloids (annuanines A and B) from the leaves and stems of *A. annua*, along with the known alkaloid fabianine. The absolute configuration of annuanine A was confirmed by X-ray diffraction. Given the very rare chemical skeleton of these compounds, we proposed a biogenetic pathway indicating that these cadinapyridine alkaloids may originate from the degradation processes of artemisinin, highlighting the dynamic chemistry of *A. annua* during storage and preparation. Despite the structural originality of the isolated molecules, the biological evaluation showed that neither annuanine A nor B exhibited notable antimalarial activity against *Plasmodium falciparum* or cytotoxicity toward cancerous or healthy cell lines, raising questions about their pharmacological relevance.

## Supporting information

**S1 File. Positive ion MS and MS/MS spectra of compound 1. Fig S2 $^1$H NMR spectrum of compound 1 (500 MHz, CDCl$_3$). Fig S3 $^{13}$C NMR spectrum of compound 1 (125 MHz, CDCl$_3$). Fig S4 HSQC NMR spectrum of compound 1 (500 MHz, CDCl$_3$). Fig S5 HMBC NMR spectrum of compound 1 (500 MHz, CDCl$_3$). Fig S6 COSY NMR spectrum of compound 1 (500 MHz, CDCl$_3$). Fig S7 NOESY NMR spectrum of compound 1 (500 MHz, CDCl$_3$). Positive ion MS and MS/MS spectra of compound 2. Fig S9 $^1$H NMR spectrum of compound 2 (500 MHz, CDCl$_3$). Fig S10 $^{13}$C NMR spectrum of compound 2 (125 MHz, CDCl$_3$). Fig S11 HSQC NMR spectrum of compound 2 (500 MHz, CDCl$_3$). Fig S12 HMBC NMR spectrum of compound 2 (500 MHz, CDCl$_3$). Fig S13 COSY NMR spectrum of compound 2 (500 MHz, CDCl$_3$). Fig S14 NOESY NMR spectrum of compound 2 (500 MHz, CDCl$_3$). Fig S15 Positive ion MS and MS/MS spectra of compound 3. Fig S16 $^1$H NMR spectrum of compound 3 (500 MHz, CDCl$_3$). Fig S17 $^{13}$C NMR spectrum of compound 3 (125 MHz, CDCl$_3$). Fig S18 HSQC NMR spectrum of compound 3 (500 MHz, CDCl$_3$). Fig S19 HMBC NMR spectrum of compound 3 (500 MHz, CDCl$_3$). Fig S20 COSY NMR spectrum of compound 3 (500 MHz, CDCl$_3$). Fig S21 Comparison of the UPLC-HRMS chromatograms obtained for the *A. annua* tea (blue) and compound 2 (red). Table S1 IC$_{50}$ (*P. falciparum*) and CC$_{50}$ (other cells) of annuanine A, annuanine B, and positive controls, in m M. Fig S22 Experimental graphs used for the determination of IC$_{50}$ of annuanines A and B and controls against *P. falciparum* FcB1 (the concentrations on the X-axis are in m g/mL for annuanines A and B and in ng/mL for artemisinin and chloroquine). Fig S23 Whole LC-HRMS chromatogram (basepeak) of *A. annua* tea prepared with fresh plant, plant dried 10 days at ambient temperature, and plant dried ten days at 45°C. Peak a is the artemisinin peak. No signal corresponding to compound 2 (Rt 4.3 min) can be seen on the chromatograms, because of the low quantity of this compound. Fig S24 Extracted ion (*m/z* 234.1-234.2 = annuanine B) LC-HRMS chromatograms of *A. annua* tea prepared with fresh plant, plant dried 10 days at ambient temperature, and plant dried ten days at 45°C. Peak b corresponds to the signal of annuanine B.**
(DOCX)

## Acknowledgments

Prof Hounnankpon Yedomonhan, head of the herbarium of University Abomey-Calavi, Benin, is greatly acknowledged for the identification of *Artemisia annua*. The authors are grateful to the producer of *A. annua*, in the plantation of Banigbe, who allowed plants to be collected from its field for this study. We are indebted to the 'Toulouse Chemistry Institute' (UAR 2599) for facilitating the recording of NMR analysis.

## Author contributions

**Conceptualization:** Nicolas Fabre, Valérie Jullian.

**Data curation:** Adrien Vitrai, Sandra Bourgeade-Delmas, Nathalie Saffon-Merceron.

**Formal analysis:** Nicolas Fabre, Nathalie Saffon-Merceron, Agnès Aubouy.

**Funding acquisition:** Nicolas Fabre, Agnès Aubouy, Valérie Jullian.

**Investigation:** Nicolas Fabre, Adrien Vitrai, Sandra Bourgeade-Delmas, Nathalie Saffon-Merceron, Abdou Madjid Olatoundé Amoussa.

**Methodology:** Nicolas Fabre, Adrien Vitrai, Valérie Jullian.

**Project administration:** Nicolas Fabre, Valérie Jullian.

**Resources:** Abdou Madjid Olatoundé Amoussa, Latifou Lagnika.

**Supervision:** Nicolas Fabre.

**Validation:** Nicolas Fabre, Adrien Vitrai, Sandra Bourgeade-Delmas, Nathalie Saffon-Merceron, Latifou Lagnika.

**Writing – original draft:** Nicolas Fabre, Valérie Jullian.

**Writing – review & editing:** Adrien Vitrai.

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
