## [Decision Letter · Decision Letter 0]

4 Jun 2025

Dear Dr. Fabre,

Thank you for submitting your manuscript to PLOS ONE. After careful consideration, we feel that it has merit but does not fully meet PLOS ONE’s publication criteria as it currently stands. Therefore, we invite you to submit a revised version of the manuscript that addresses the points raised during the review process.

We look forward to receiving your revised manuscript.

Kind regards,

Timothy Omara

Academic Editor

PLOS ONE

Additional Editor Comments 

Dear authors,

Reviewers have commented on your submission, and as you will notice, there are quite a number of concerns that needs to be addressed before the submission can be reconsidered for publication. In addition,

1. I recommend that you revise the title to indicate the bioactivities tested. It is a very common scientific practice to guide the reader on the content of the manuscript.

2. The abstract shows that you utilized MS yet UHPLC-HRMS seems to have been used. Please, clearly indicate which spectroscopic analyses were conducted, and in the best case which 1D- and 2D-NMR analyses were performed.

3. You indicated that the samples were from both Benin and France. But it turns out it is only from Benin.

4. Other comments are in the attached MS file

Reviewers' comments:

Reviewer's Responses to Questions

**Comments to the Author**

1. Is the manuscript technically sound, and do the data support the conclusions?

Reviewer #1: No

Reviewer #2: Yes

Reviewer #3: Partly

2. Has the statistical analysis been performed appropriately and rigorously?

Reviewer #1: N/A

Reviewer #2: N/A

Reviewer #3: N/A

3. Have the authors made all data underlying the findings in their manuscript fully available?

Reviewer #1: No

Reviewer #2: Yes

Reviewer #3: Yes

4. Is the manuscript presented in an intelligible fashion and written in standard English?

Reviewer #1: No

Reviewer #2: Yes

Reviewer #3: No

Reviewer #1: Generally, the article describes new structures (molecules found in A.

annua). Although they did not show antiplasmodial activity neither

activity against abnormal cells (perhaps cancerous cell), they could

contribute on the chemistry of A. annua.

However, the manuscript is poorly written, the biological tests

(antiplasmodial, anticancer, MTT assay) are poorly described with lack

of references in most cases. The authors talk about A. annua teas

throughput but the compounds isolated is not from the teas. Theres no

coherence, lack of consistency in the methodology and results. The

conclusion is misleading (not clear).

The results section is poorly described with no information on the

biological test.

I therefore recommend the manuscript to be reformatted, and rewritten

to focus on phytochemistry of A. annua and lay emphasis on the new

isolated molecules.

Reviewer #2: This is a well written manuscript but take note of the attached recommendations and work on them according. This done make some good discussions in the NMR data. The detailed discusion is imbeded within the manuscript and some other attached responses.

Reviewer #3: Title: Cadinapyridine sesquiterpene alkaloids from Artemisia annua L.

Abstract: The authors need to name the new compounds 1 and 2 in the abstract, since this is their main finding.

Introduction: The introduction starts well with the description of the plant, its uses, the studies that have been done on the plant and the description of the identified phytochemicals. Then the chapter ends with a paragraph that highlights the aim, methods and partial findings, which sounds more of an abstract.

Comment: this chapter misses the research gap, and the specific research questions to be addressed.

Materials and Methods

-Line 85: The subheading “General materials” is misleading, since it doesn’t seem related to what is under it. Kindly give the appropriate heading, and then shift it to be after the section “Extraction and purification of alkaloids”. Because, purification comes after Extraction. See line 86 says “Compounds were purified on silica cartridges...” You can’t purify before extraction please. Have some logical flow of ideas.

-Kindly maintain the same tense under this section. There is mixed use of “was” and “is”

-Line 132: How did you identify Fabianine (F1-4, compound 3) at this level, before LC-MS and NMR. Much as it is a known compound, it was not identified at this point. So, I would suggest just putting “compound 3 (F1-4),

-Line 138. This coding of subfractions is confusing “compound 1 (F4-1 - 2 mg)” It could mean subfraction 1 of fraction 4 or whatever... What would it mean for someone who considers it as fraction F4-1 -2?? And then gets stuck at the meaning of mg? Does that become the problem of the reader or the writer? Check that all the codes are defined to avoid confusion.

-Lines 147 to 156 look like results to me, yet there is a section of results. I would request the authors to have logical flow of information/ideas

Results

-Though this section highlights results, some discussions of the results appear here, only that the references are lacking. Phrases like “...indicating six degrees of unsaturation (Line 258)”, “...suggesting a primary amine (loss of NH3) in the structure of 1. (Line 260)”, “...indicating the presence of three methyl groups, two methylene groups, five methines (including two aromatics), and four quaternary carbons... (Lines 263 down wards...)”, “...were in good agreement with those of Fabianine [18], an alkaloid first isolated from Fabiana imbricata by Edwards and Elmore in 1962. [19] Therefore,... (Line 335 down wards...”

-How can a section of results have references even?

-I would think, interpretation of the findings or adding meaning to them is discussion.

-The authors need to cite the literature that guided the MS and NMR assignments

NB: The authors need to know what goes where and in which section.

Conclusion:

-The authors lack the research gap and their questions are not articulated. Thus, it is very hard to assess their conclusion, to see if it is matching

**Do you want your identity to be public for this peer review?** For information about this choice, including consent withdrawal, please see our Privacy Policy

Reviewer #1: No

Reviewer #2: **Yes: ** Ivan Kiganda (PhD)

Reviewer #3: No

---

## [Author Response · Author response to Decision Letter 1]

18 Jul 2025

The response to editor and reviewers is included in the .docx file titled "response to reviewers"

---

## [Decision Letter · Decision Letter 1]

13 Aug 2025

Cadinapyridine sesquiterpene alkaloids from Artemisia annua and in vitro cytotoxicity and antiplasmodial activities

PONE-D-25-22406R1

Dear Dr. Fabre,

We’re pleased to inform you that your manuscript has been judged scientifically suitable for publication and will be formally accepted for publication once it meets all outstanding technical requirements.

Kind regards,

Timothy Omara

Academic Editor

PLOS ONE

Additional Editor Comments (optional):

Reviewers' comments:

Reviewer's Responses to Questions

**Comments to the Author**

Reviewer #2: All comments have been addressed

Reviewer #3: All comments have been addressed

2. Is the manuscript technically sound, and do the data support the conclusions?

Reviewer #2: Yes

Reviewer #3: Yes

3. Has the statistical analysis been performed appropriately and rigorously?

Reviewer #2: N/A

Reviewer #3: N/A

4. Have the authors made all data underlying the findings in their manuscript fully available?

Reviewer #2: Yes

Reviewer #3: Yes

5. Is the manuscript presented in an intelligible fashion and written in standard English?

Reviewer #2: Yes

Reviewer #3: Yes

Reviewer #2: Based on my review, O accept the manuscript ro be published as the author improved the write up and also attended to almost all the queries we raised.

Reviewer #3: The comments have been sufficiently addressed. I therefore suggest that the paper be accepted for publication.

**Do you want your identity to be public for this peer review?** For information about this choice, including consent withdrawal, please see our Privacy Policy

Reviewer #2: **Yes: ** Ivan Kiganda

Reviewer #3: No

---

## [Editor Report · Acceptance letter]

PONE-D-25-22406R1

PLOS ONE

Dear Dr. Fabre,

I'm pleased to inform you that your manuscript has been deemed suitable for publication in PLOS ONE. Congratulations! Your manuscript is now being handed over to our production team.

Kind regards,

on behalf of

Dr. Timothy Omara

Academic Editor

PLOS ONE